# The role of mainstreamness and interdisciplinarity for the relevance of scientific papers

Stefan Thurner[1,2,3,4]*, Wenyuan Liu[5], Peter Klimek[1,2], Siew Ann Cheong[5]

**1** Section for Science of Complex Systems, CeMSIIS, Medical University of Vienna, Spitalgasse, Austria, **2** Complexity Science Hub Vienna, Josefstädter Strasse, Vienna, Austria, **3** Santa Fe Institute, Santa Fe, NM, United States of America, **4** IIASA, Schlossplatz, Laxenburg, Austria, **5** Division of Physics & Applied Physics, School of Physical and Mathematical Sciences, Nanyang Technological University, Nanyang Link, Singapore

* stefan.thurner@meduniwien.ac.at

**Data Availability Statement:** The data underlying the results presented in the study are owned by and available upon request from American Physics Society (data-requests@aps.org). Interested researchers must request access on their webpage (https://journals.aps.org/datasets).

## Abstract

Is it possible to tell how interdisciplinary and out-of-the-box scientific papers are, or which papers are mainstream? Here we use the bibliographic coupling network, derived from all physics papers that were published in the Physical Review journals in the past century, to try to identify them as mainstream, out-of-the-box, or interdisciplinary. We show that the network clusters into scientific fields. The position of individual papers with respect to these clusters allows us to estimate their degree of mainstreamness or interdisciplinarity. We show that over the past decades the fraction of mainstream papers increases, the fraction of out-of-the-box decreases, and the fraction of interdisciplinary papers remains constant. Studying the rewards of papers, we find that in terms of absolute citations, both, mainstream and interdisciplinary papers are rewarded. In the long run, mainstream papers perform less than interdisciplinary ones in terms of citation rates. We conclude that to avoid a unilateral trend towards mainstreamness a new incentive scheme is necessary.

## Introduction

Science has become a tremendously expensive industry over the past century. The world's current total nominal Research and Development spending is approximately two trillion US dollars [1]. The amount of publications has increased exponentially for more than a century. Scientific output measured in numbers of papers has increased from about 2000 in 1900 to one million papers in 2010 (Web of Science). In physics alone, in the same timespan papers rose from about 200 to 200,000 [2]. There are signs, however, that science might become less efficient and that its output in terms of groundbreaking discoveries and inventions—not the number of papers published or PhDs granted—is declining. In 1996, Leo Kadanoff stated "The truth is, there is nothing—there is nothing—of the same order of magnitude as the accomplishments of the invention of quantum mechanics or of the double helix or of relativity. Just nothing like that has happened in the last few decades." [3]. In a more recent study, a similar conclusion is drawn in a survey of leading scientists in various fields based on their opinion on

**Funding:** We acknowledge support from the Singapore Ministry of Education Academic Research Fund under grant number MOE2017-T2-2-075 to SAC and from the Austrian FFG Project 857136 to ST. The funders had no role in study design, data collection and analysis, decision to publish, or preparation of the manuscript.

**Competing interests:** The authors have declared that no competing interests exist.

relevant contributions to science over the past century [4]. There are various possibilities to explain a possible decline of rates for fundamental scientific discoveries. Either most of the discoverable things have been discovered already. This view that can be tremendously wrong, as we know for example from a dubious statement by Lord Kelvin in 1900, "There is nothing new to be discovered in physics now. All that remains is more and more precise measurement." [5]. Or, alternatively, the quality of scientists is going down, or the appetite and incentives for solving new and big problems with new and risky frameworks is declining.

When choosing a scientific problem, a scientist can choose a big problem that no one was able to solve before—most likely because a methodological framework or the technological means are not yet there—or a small one that only incrementally improves upon generally accepted knowledge, and for which an accepted framework, technology, and an informed community already exists. Doing innovative science often means not only to step out-of-the-box and think anew, invent novel and adequate frameworks, views and eventually solutions, but also—in case of success—one has to fight the community and the keepers of current dogmas to accept new ways of thinking [6]. This is risky and—even though often beneficial to science—can be detrimental to scientific careers. Indeed, most scientists seem to opt for the low-risk option. In [7] it was found that the vast majority of papers in biomedicine and chemistry published between 1934 and 2008 were building on existing knowledge rather than generating novel and innovative findings. They attribute their findings to an inadequate incentive structure with a publish-or-perish philosophy that hinders innovation and selects for a timid science that guarantees sure citations in a predictable, and also timid, environment or community. Scientists choosing the low-risk option conclude that innovative research is a suboptimal way to gain scientific recognition, a "gamble whose payoff, on average, does not justify the risk of not getting published" [7]. It is sometimes argued that "timid science" is needed to solidify and reproduce novel findings and to create a broad base out of which truly innovative science can emerge. However, there are indications that these roles are not carried out properly, how could it otherwise be possible that a tremendous fraction of papers, even in top journals, cannot be reproduced, regardless of discipline [8, 9]. For an overview of the extent of the replication crisis, see e.g. [10].

The prevalent incentive scheme in science production is based on productivity factors, such as numbers of papers, quality factors, such as citations, and cumulative indicators, like the h-factor and its variations. These indicators create the questionable belief that people without knowledge in science can make decisions such as which scientists should be hired or funded. This is maybe true for incremental science but certainly not for judging, who is creative enough and has the potential, strength, and courage to carry through true breakthroughs that move knowledge forward. These indicators pose incentives to produce papers that stay close to the mainstream. The mainstream—by definition—contains the largest pool of scientists that can cite you. Papers receive more citations than others published on the same topic at the same time, if their abstract simply uses keywords occurring in a larger number of other abstracts [11]. Most scientists know the mainstream literature well. Incremental mainstream ideas will face less resistance than novel ideas that are hard to understand and might contradict and surpass the present standard of the community. In today's scheme it is better to hire a post doc that produces a predictable number of papers at a certain quality level than to "feed someone through for a decade" with the risk of not having a single paper at the end, and to be rated as a loser team. Examples like these indicate that it is rational for scientists to publish in the mainstream, given that they value their careers more than they love the pure progress of science.

Can increased scientific competition and more top-down management with the aim to increase the fraction of high-risk/high-gain science improve the situation? Maybe not, as a

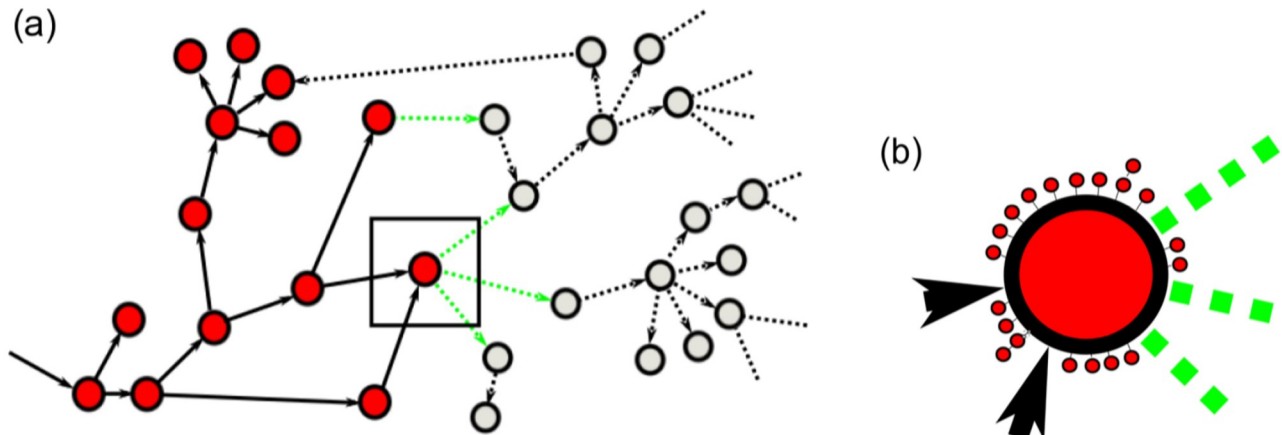

**Fig 1. Schematic cartoon picture of the way science progresses.** (a) Every node represents a paper that makes a significant contribution to science. Red nodes are published past discoveries; grey nodes are yet undiscovered (but discoverable) scientific facts. Black arrows indicate which discovery (or paper) influenced which. The set of dashed green lines is the "adjacent possible", i.e. the set of scientific facts that can be discovered, given the present state of knowledge (the set of red nodes). Dashed grey lines show novel opportunities that open up once progress has been made. (b) The blow up shows what happens around a significant contribution: many incremental papers (small nodes) repeat, confirm, validate, and explore the "vicinity" of what was found in the breakthrough paper (big red node). These small nodes constitute the mainstream. By definition, incremental papers do not make much headway towards new big discoveries.

recent study suggests; competition and management do not seem to improve science output [12]. The present incentive scheme does not seem to reward risky science, as it has the tendency to select the mainstream, see also [13].

Science progresses discovery by discovery. Usually discoveries are presented in papers. Not every paper is a discovery; mainstream science papers often are not. Typically, new innovations build on existing knowledge, often novelty arises from new re-combinations of existing knowledge and ideas. The case in science is similar to progress of technology [14]. Fig 1a shows a cartoon image of the "progress of science". Every node represents a paper that made a significant contribution to science (innovation or breakthrough). Red nodes are discoveries made in the past that have been published. Grey nodes are hitherto undiscovered—but discoverable—scientific facts. Black arrows indicate which work influenced which. The set of dashed green lines is the so-called "adjacent possible", the set of scientific facts that can be discovered within the next time period, given the state of current knowledge, i.e. the set of red nodes. Dashed grey lines show the possibilities that open up once new progress has been made. Once a discovery is made, the corresponding grey node turns into a red one. Incremental or timid mainstream research is depicted in Fig 1b. It shows a blow up of Fig 1a.

In this paper, we want to find out if this picture is correct and can be supported by data. In particular, we ask how out-of-the-box and bridging science is rewarded in terms of citations in the long run compared to mainstream. We use several measures to estimate the degree of "interdisciplinarity" of individual papers. For this, we use the bibliographic coupling (BC) network [15] of all papers that appeared in one of the Physical Review journals in the last century. It is a way to quantify the similarity of papers. In the BC network, $M$, papers are represented as nodes, a link is defined between two papers $i$ and $j$ if they both cite a common paper. The weight of the link, $M_{ij}$, is the overlap of the reference lists of the two citing papers. The BC network can be seen as a rough proxy for the picture shown in Fig 1, in particular for the existing red nodes. BC networks clearly exhibit clusters of similar scientific areas or fields, as papers within closely related areas are linked with each other through the same references, see Fig 2. This reflects the fact that authors that constitute a discipline tend to read the same literature.

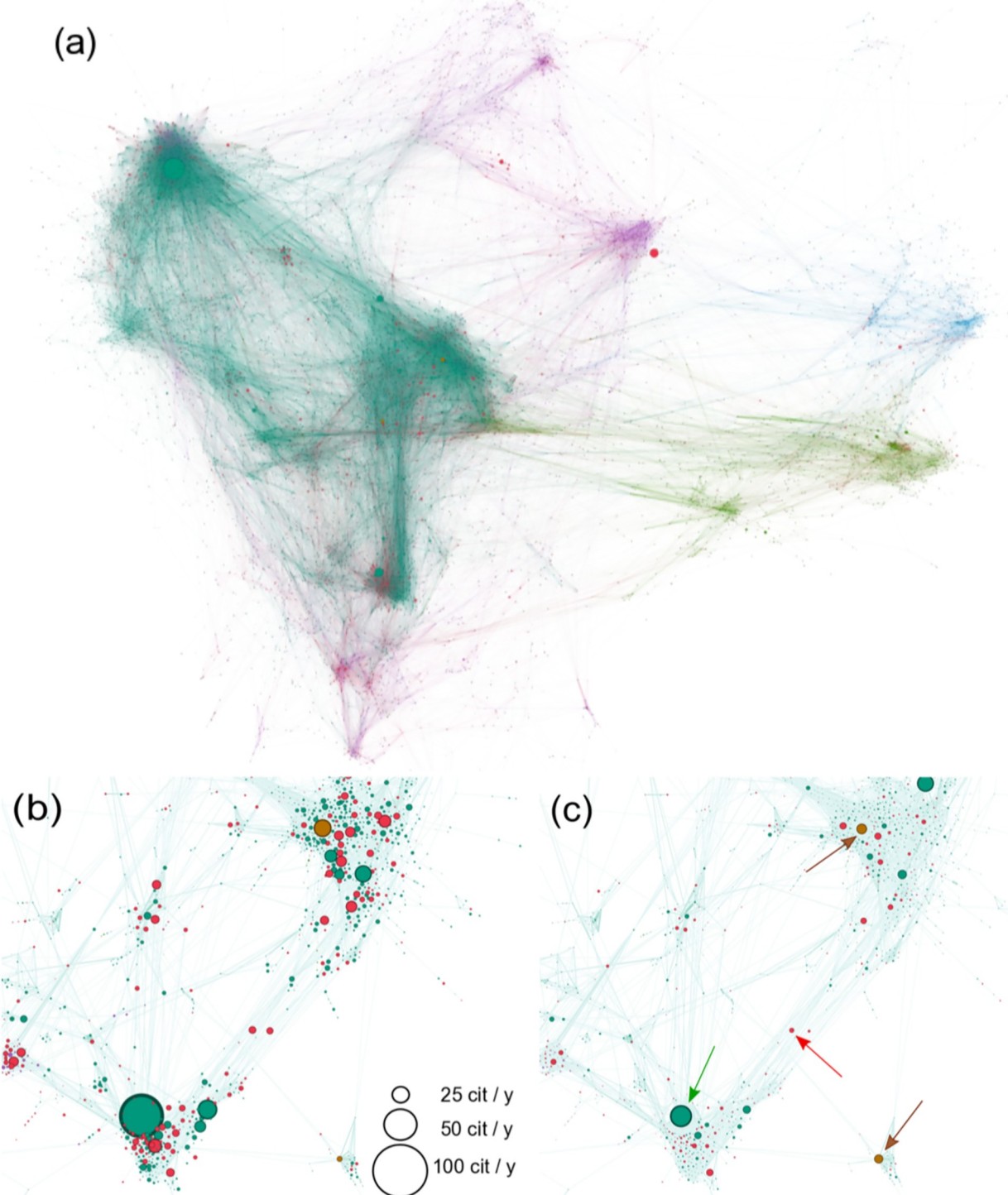

**Fig 2.** (a) Bibliographic coupling network of the 8673 Physical Review papers published in 1991, PRA purple, PRB turquoise, PRC blue, PRD green, PR Letters red, Reviews of Modern Physics brown. Clusters of all sizes are present. Node size represents the number of citations of these papers in 2011. Clusters are clearly linked by bridging papers. (b) Section of the network, thresholded to link weights two and larger. Node size represents the *citation rate* during a two-year period after publication, $C_i^2/2$. Many of the immediate citations go to papers close to the cluster centers. (c) Same section for citation rates over a twenty-year period after publication, $C_i^{20}/20$. For explicit examples and the meaning of the arrows, see S1 Text. Overall, citation rates over twenty years are smaller than for the first two years. Many papers that are relevant on the long timescale appear in the periphery of clusters (out-of-the-box) and between clusters. Most papers close to cluster centers become marginal on the long timescale. Note the positions of PRL papers (red) and review articles (brown). PRL papers attract much short-term attention but are no longer dominant in the long run. One of the two Rev. Mod. Phys. articles appears in the periphery of a big cluster, the other represents a small emergent field.

Interdisciplinary papers often "link" works from different areas. In this sense, Einstein's 1905 relativity paper would bridge the areas of mechanics and electrodynamics. To quantify interdisciplinarity, we take two approaches. First, we use the minimal distance of a paper to the center of the nearest cluster. Clusters we compute by k-means clustering, see Methods. Mainstream papers would appear near the cluster centers, which are an output of the k-means algorithm. The periphery can then be defined as regions with a sufficiently large distance to the cluster centers. As a quantity for reward and a proxy for relevance of a paper in the long run, we look at the number of citations it acquired two ($C_i^2$) and twenty years ($C_i^{20}$) after its publication. We hypothesize that two things should be observable: (i) the existence of many immediately well-performing papers in the cluster centers, and (ii) an over-representation of well-performing papers located at the periphery and between clusters at a later stage. These need some time to be discovered and understood, and should not appear immediately, but only after some time. For a second measure of interdisciplinarity, we follow an idea reported in [16], based on the Physics and Astronomy Classification Scheme (PACS) numbers that are used by authors to assign research areas to their papers. Typically, more than one PACS number is used. The interdisciplinarity of a paper is associated with the diversity of the PACS numbers of its references. The diversity of every paper $i$ is measured by its "PACS entropy", $I_i$, see Methods.

It is not the purpose of this paper to predict scientific success of papers and scientists. This has been done in recent works [2, 17–19]. Here we use the BC network to identify papers as mainstream, out-of-the-box, or interdisciplinary. This is different to recent work where the role of interdisciplinary science has been explored in terms of self-reported classification schemes [20]. The present approach allows us to map the position of papers more precisely and thereby contribute a number of complementary findings. We want to elucidate the positions of "important" papers within that network and show that these often are indeed out-of-the-box and interdisciplinary, especially at longer timescales. Note that we are in no way making judgements about whether incremental or interdisciplinary are more relevant. They perform different roles, the first pushes the boundary the latter consolidates and confirms or rejects. Both are necessary for the advancement of science.

## Results

In Fig 2a we show the BC network of the 8673 papers published in 1991 in all of the Physical Review journals (PR A, PR B, PR C, PR D, PR Letters, and Reviews of Modern Physics). Small and large clusters of research areas are clearly distinguishable, ranging from a few to hundreds of papers. Node colors mark different journals. Node size is the number of citations in 2011, $C_i^{20}$. In this network, interdisciplinary or bridging papers would be positioned between clusters; "out-of-the-box" papers would be found in the periphery of clusters. Mainstream papers are typically in the center of clusters. Papers of all this type are visible in Fig 2b and 2c. Here node size is the annual *citation rate* measured two years after publication in (b), $C_i^2/2$, and after twenty years, $C_i^{20}/20$, in (c). We show rates to be able to sensibly compare rewards at the two time scales in (b) and (c). Obviously, annual citation rates observed over 20 years are smaller than when measured in the first two years after publication. It is visible by plain inspection that many well-cited papers on the long timescale, (c), appear in the periphery of clusters (out-of-the-box) and between the clusters (bridging). Papers in the cluster centers seem to become relatively more marginal in the long term. Note the positions of PRL papers (red) and the review articles (brown). PRL papers seem to attract much short-term attention but cease to be dominant in the long run. One of the two review articles (brown) appears in the periphery of a big cluster [21]. It states a sentence in the abstract that clearly marks it as a paper linking

various fields: "[...] and rather conventional picture emerges from a number of techniques–analytical (spin-wave theory, Schwinger boson mean-field theory, renormalization-group calculations), semianalytical (variational theory, series expansions), and numerical (quantum Monte Carlo, exact diagonalization, etc.)." The other review article represents an emerging bridging field [22]. Its title captures its non-mainstreamness: "Phenomenological theory of unconventional superconductivity". For more details, see S1 Text.

## Temporal trends

We next look at historical trends of where papers are localized in the BC network. In Fig 3a we see the distribution of the distances to the nearest k-means cluster, $D_i$, for the years 1981, 1991, and 2001. Over the three years the distribution shifts to the left; the medians change from $2.25 * 10^{-3}$ in 1981, to $1.96 * 10^{-3}$ and $1.84 * 10^{-3}$, in 1991 and 2001, respectively. A Wilcoxon rank sum test for equal medians yields p-values $< 10^{-9}$ for all possible pairs of years. The tail of the distribution is similar for the three years. This means that there is a tendency of papers shifting towards the cluster centers, at the expense of the fraction of papers that sit at the periphery; there is practically no change in the fraction of papers between clusters. The tendency that clusters get more populated in the center is also seen in the degree distributions of papers in the BC network. The distribution functions for the same three years are shown in Fig 3b. The distribution changes toward higher degrees; medians shift from 16 in 1981 to 26 and 41 in 1991, and 2001, respectively. The Wilcoxon rank sum test yields highly significant p-values $< 10^{-81}$ for all pairs of years. Papers get more similar to many others. Both plots indicate that over time, clusters become more populated in the centers and that the relative contribution of bridging papers does not change over time.

## Conditional distributions of citations

Fig 3 shows the 50%, 70%, and 90% quantiles of the distribution of the 20 year citations, $C_i^{20}$ in 1991, conditioned on the minimal distance to the nearest cluster, $D_i$, see (c), and conditioned on the degree of the papers, see (d). The 50% quantile is the median. We partitioned the data along distance and degree into bins that contain 400 data points each. In this way, a reasonable definition of the quantiles along distance and degree is possible. In both cases, (c) and (d), it is visible that the median (blue), the 70% (red), and the 90% (green) quantiles rise significantly with distance and degree. This means that two effects take place simultaneously: first, out-of-the-box and interdisciplinary papers seem to be rewarded (large distances) and, second, as one would expect, mainstream publications are rewarded in terms of citations. Not surprisingly, the more papers a given paper is linked to (degree) it is cited. We verified that if citations are assigned to randomly chosen papers, constant quantiles at the appropriate levels are obtained.

In Fig 3e we present the distribution function of twenty-year citations for short distances to the nearest cluster (red), and for large ones (blue). The distribution for short distances (in the leftmost bin in (c)) contains all papers with a distance in the range of $D_i \in [0, 2.3 * 10^{-4}]$. Large distances (rightmost bin in (c)) cover the data in the range of $D_i \in [9.2 * 10^{-3}, 1.27 * 10^{-2}]$. The citation distribution changes visibly towards larger medians, from 3 to 13 (Wilkoxon test $p < 1.68 * 10^{-56}$). The same type of citation distribution is shown in Fig 3f for small (red) and large (blue) values of the degree. The same pattern is found: for high degree papers, the citation distribution has a higher median (Wilkoxon test $p < 2.01 * 10^{-35}$). We find similar results also for the betweenness, see Figs Ba and Bb in S1 Text. however, somewhat less pronounced than for distances. Closeness, $K_i$, of papers as defined as the inverse of the average (network) distance to any other node, see Methods, again shows similar behavior, see Figs Bc and Bd in

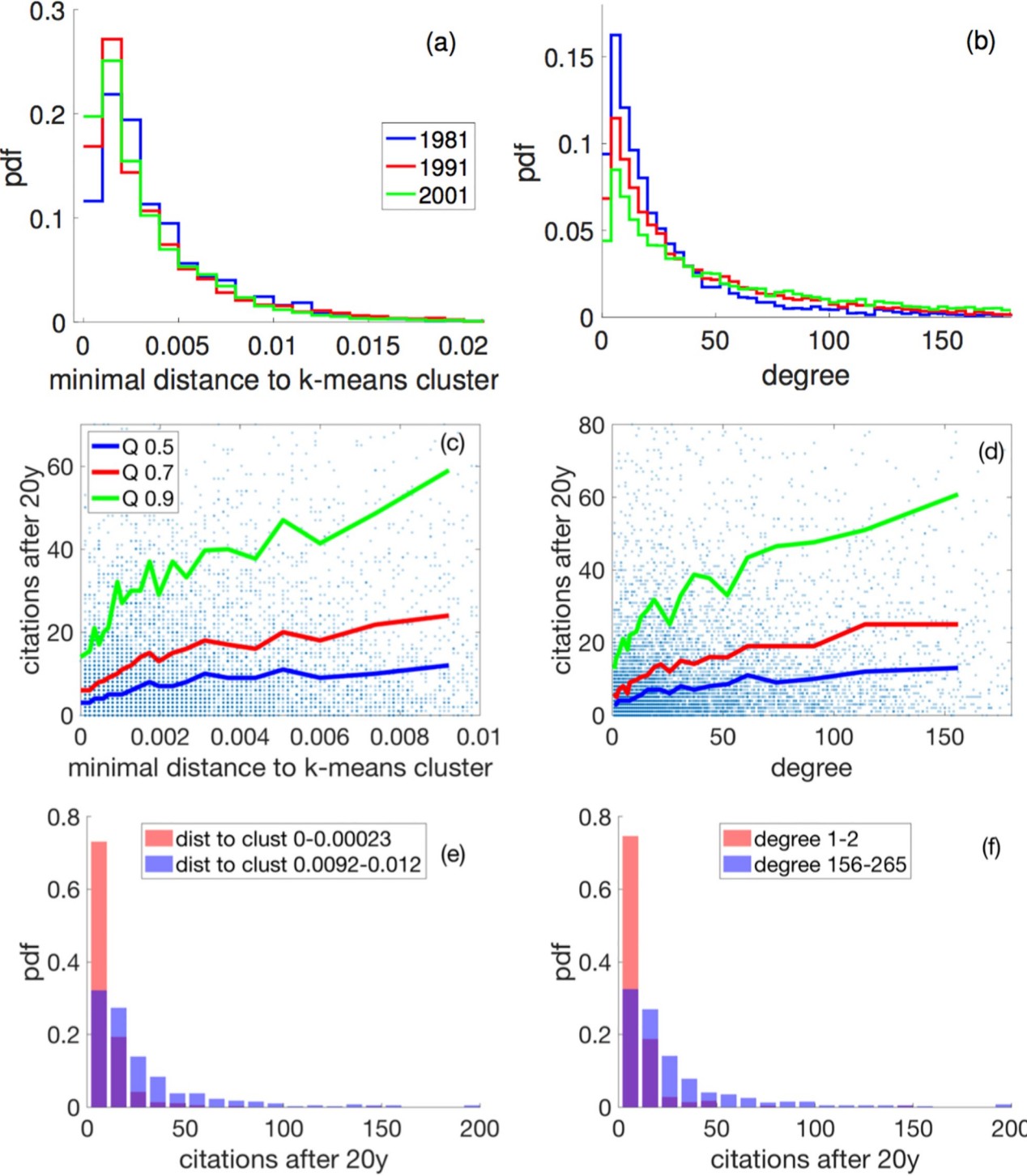

**Fig 3.** (a) Distribution of the distances of individual papers to the centers of their nearest clusters in the years 1981 (blue), 1991 (red), and 2011 (green). Over time, the distribution shifts toward smaller distances, i.e. more papers tend to appear in cluster centers. (b) The distribution of degrees over the same years shift towards much larger values (tail increases), i.e. there is a tendency to increasingly link to more similar papers. (c) Scatterplot of citations of papers published in 1991, twenty years after their publication, $C_i^{20}$, versus their distance to cluster centers. The 90%, 70%, and 50% quantiles are shown in green, red, and blue, respectively. Citations increase with higher distances from clusters; bridging papers are awarded in the long run. (d) Citations, $C_i^{20}$, versus their degree. A clear increase is apparent. (e) Distribution of citations for small and large values of distance. The plot is a normalized histogram of the 400 papers with the shortest distances. The blue distribution is for the 400 papers with the largest distance. (f) Distribution of citations for small and large degree.

 For completeness, we further checked the dependence of citations of paper $i$, $C_i^{20}$, on the respective length of its reference list, $L_i$. We see a small effect, Figs Be and Bf in .

## PACS diversity

In  we show the central section of the scatterplot of the citations after 20 years, $C_i^{20}$, against the PACS entropy, $I_i$; for its definition, see Methods. We show the 0.9, 0.7, and the 0.5 (median) quantiles of the citation distributions measured in bins that contain 400 papers each. All quantiles increase by a factor of more than two. The distribution function for the citations for the range of PACS entropy, $I \in [0.10–0.19]$, is drawn in  in red, for the range, $I \in [0.30–0.31]$, in blue. We find a strong correlation between PACS entropy and the length of reference list ($\rho = 0.61$), in , from which a linear relation of the median (blue) can be inferred. To naively control for the length of the reference list, we show the scatter plot of $C_i^{20}$ versus the PACS entropy divided by the reference list, $I_i/L_i$, see . The effect reverses and quantiles decline, showing that the explanatory power of the PACS entropy might be strongly confounded by the number of references; see also regression analysis below and in .

## From papers to authors

Do these findings also hold for authors? By associating papers to authors we observe similar results. In  we show the citations of authors versus the same network measures as in . To this end, we identify all papers of authors that were published in the period 1981-1991. We count all citations of all of these papers up to 2011. For every year between 1981-1991, we construct the BC network and compute the average distance to nearest clusters, the average degree, and the average betweenness for all the papers of that author in that year. We finally average over all years 1981-1991 for all authors.  shows scatterplot and quantiles for author citations versus the average nearest distances. In  the corresponding distributions for small (red) and large (blue) distances are shown. Medians shift from 3.5 to 9.0 (Wilkoxon $p < 10^{-202}$).  display the situation for the degree. For small and large values the medians of the respective distributions increase from 3.0 to 12.3 (Wilkoxon $p < 10^{-300}$). For the betweenness, seen in (e)-(f), medians for small and large values shift from 4.5 to 9.8 (Wilkoxon $p < 4.35 * 10^{-5}$). The results for the short-term citations of authors can be seen in Fig C in , where the cumulative combined citations in 1993 of all papers produced between 1981-1991 are shown; same panels as in .

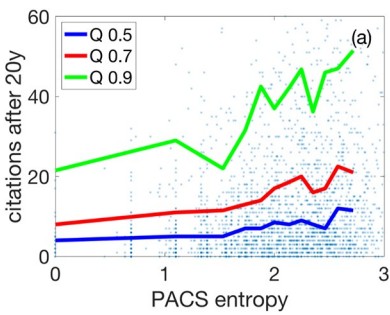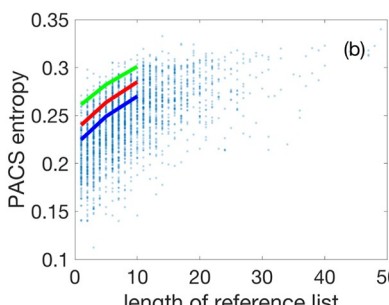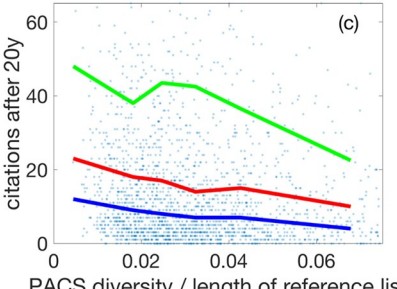

**Fig 4.** (a) Scatterplot of 20-year citations, $C_i^{20}$, versus their PACS entropy, $I_i$. To control for the strong correlation between $I_i$ and the length of reference list, $L_i$, ($\rho = 61$), see (b), we show the PACS entropy per reference, $I_i/L_i$, in (c). The effect seen in (a) has vanished and is reversed. Only those 2, 491 papers where enough PACS information is present were considered, see Methods.

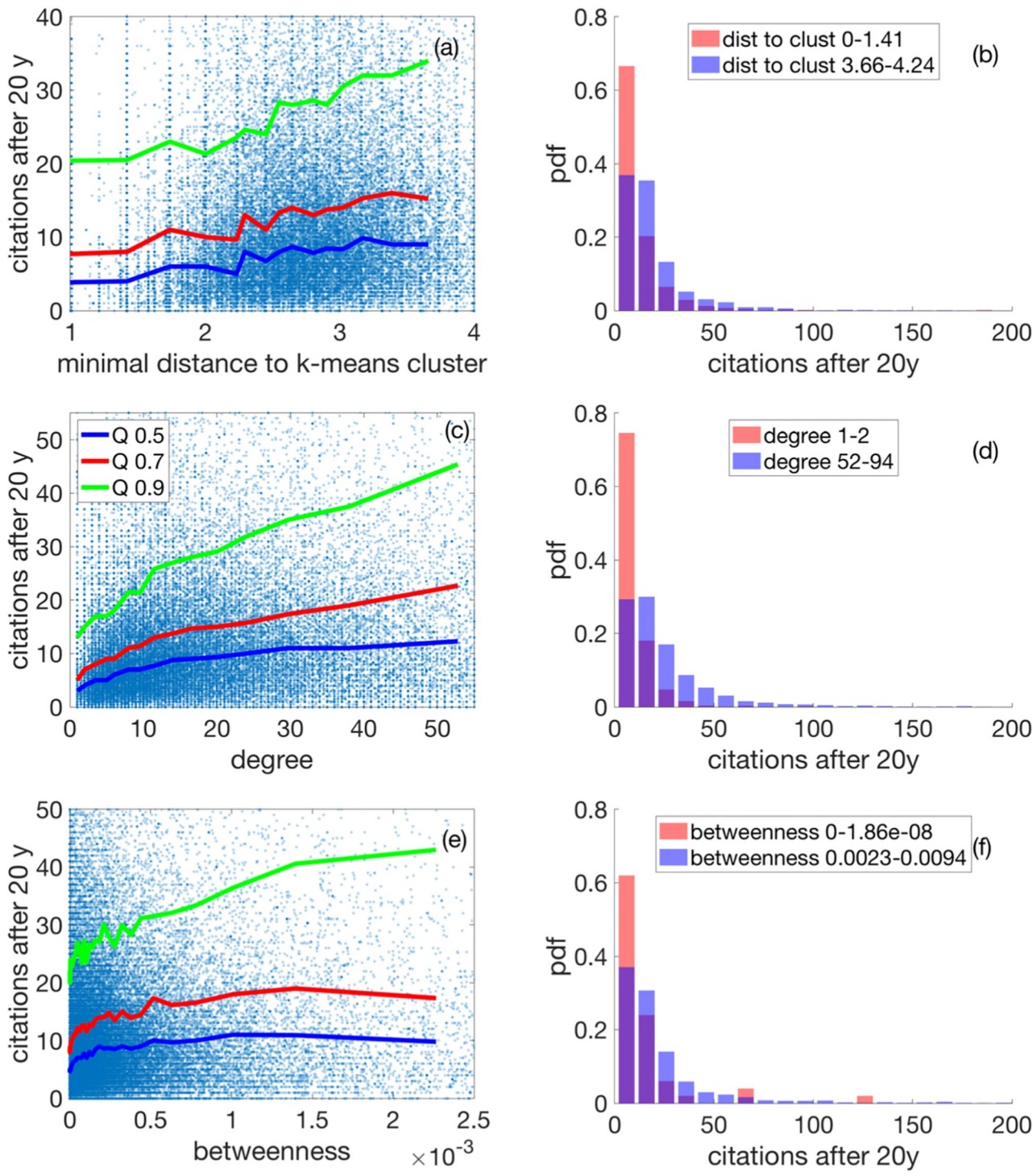

**Fig 5. Dependence of citations of authors on network measures; dots now represent authors.** (a) Scatterplot of all citations up to 2011 of all those papers an author has published between 1981 and 1991, versus the average distance of these papers to their respective closest cluster in the BC network in the year of publication. 90%, 70%, and 50% (median) quantiles are shown in green, red, and blue, respectively. Citations increase with higher average distances. (b) Distribution of authors' citations for short (red) and large (red) distances. The plot is a normalized histogram of the 4000 authors with the smallest distance. The blue distribution is for the 4000 authors with the largest distance. (c) and (d) show the case for the degree. Again, citations increase strongly with degree. The betweenness results are seen in (e) and (f). As for papers, the effect for betweenness is weak.

## Regression analysis and robustness tests

To better understand the extent to which our results could be confounded by the length of the reference list, $L_i$, we perform a regression analysis, see S1 Text. For each considered dependent variable, we find a strongly significant positive linear relationship with citations after 20 years, $C^{20}$, see Table A in S1 Text. The strongest relations are observed for the degree, which increases with citations by a factor of 0.33(1), and for distances that increase with a factor of 0.26(1). Numbers in brackets denote standard deviations at the last significant digit. In both cases we have $p < 10^{-130}$ against the null hypothesis that the true coefficient value is zero. After adjusting for the length of the reference list, $L_i$, these relations remain strongly significant (coefficients of 0.31(1), $p < 10^{-100}$, for the degree; 0.18(1), $p < 10^{-46}$, for the distance). The correlation with the PACS entropy vanishes almost entirely (from 0.22(2), $p < 10^{-28}$, to 0.08(3), $p = 0.008$, after the adjustment), see Table A in S1 Text. Similar observations hold for author-level results; see Table B in S1 Text, where we show that the correlations of author citations with degrees, distances, closeness, and betweenness remain strongly significant after adjusting for reference list length or the number of publications.

## Discussion and conclusion

Current incentive structures almost exclusively reward the production of mainstream science. It is not only the increasing importance of the number of citations or the h-factor, it is also that papers and proposals will only be accepted if they are sufficiently understood by peers— which is often not the case for out-of-the-box and novel ideas that need backgrounds from more than one field to be understood. To suggest high-risk papers, projects, or individuals poses reputational risk for referees and committee members. Even though high-risk/high-reward science is highly needed by society it is only happening to an astonishingly low degree in academia.

Here we explored the extent to which scientific work can be quantified as mainstream, out-of-the-box, or interdisciplinary. We study the bibliographic coupling network and find that it is nicely structured into clusters of various sizes. Clusters are groups of papers that cite the same literature, i.e. constitute scientific areas. The existence of these clusters allows us to actually visualize how mainstream, out-of-the-box, or interdisciplinary a paper is by locating it in this network, relative to nearby clusters. Mainstream papers are located close to cluster centers. Bridge- and interdisciplinary papers are found between clusters. Note that interdisciplinary articles can of course be mainstream and out-of-the-box. Here we focus on the status of new and emerging papers. When we talk about interdisciplinary papers we have in mind young papers that are found in small clusters between well-established large clusters of mainstream work. We define interdisciplinarity in three ways, using PACS numbers, betweenness and the position between clusters. These definitions overlap to varying degrees. While the PACS numbers are self-assigned by the authors, the other two BC network-based definitions are more objective. The PACS numbers however allow us to independently check for the validity of the results. The other notions are strongly related.

To estimate the reward of papers we simply count their citations two and twenty years after their publication. We find that mainstream is indeed rewarded in terms of absolute numbers of short-term citations. However, this is not the case for *citation rates*, where many out-of-the-box and interdisciplinary papers do better in the long run. In the long run, citation rates near the cluster centers decline when compared to many papers on the periphery or between clusters. See changes in node sizes from Fig 2b to 2c. When looking at temporal trends, we see that the fraction of mainstream papers increases considerably from 1981 to 2001, while the fraction of interdisciplinary papers stays practically constant. This is visible in Fig 3a, where there is a

strong increase in the first bin, whereas bins 3-5 decrease from 1981 to 2001. The tail is practically unaffected, meaning that the fraction of bridging papers remains constant. The number of out-of-the-box papers decreases in favor of the mainstream papers.

The "Science of Science" is a concept introduced more than half a century ago [23]. Its recent revival manly focuses on various aspects of science production, in particular on the citation mechanism [17], impact prediction [18], or on scientific careers [24]. In [17] a mechanistic model that incorporates preferential attachment, attention decay, and "fitness" was proposed to predict the long-term citation impact based on a paper's early citation history. A study of 2887 physicists in [18] found that factors leading to highly cited papers are not random. By combining productivity and a scientist-specific "Q factor", they propose a stochastic model to explain scientific success. Analyzing data of 200 leading scientists and 100 assistant professors, [24] found that persistent career trajectories lead to increasing returns in the scientific production. The model there also shows that short-term contracts may lead to early career termination [24]. The role of early career co-authorships is studied in [19]. The importance of a mesoscopic picture on knowledge evolution was realized in [25]. There topical clusters of APS papers were analyzed and visualized across a century with alluvial diagrams. The roles of mainstreamness and interdisciplinarity have so far not received much attention, even though the topic has been identified, discussed, and even used by funding agencies [26]. An important contribution in this direction is [16] that uses the PACS diversity of authors (defined differently than here) to demonstrate that authors with very low (experts) and very high PACS diversity (very interdisciplinary) are on average cited much better than authors with intermediate PACS diversity. our results are nicely in line with important recent work that focuses on the role of interdisciplinarity [20]. The authors, focussing on the PACS numbers of papers for estimating interdisciplinarity levels find that interdisciplinarity is helpful for success in terms of citations and papers. Also in line with their results is the relatively large role of randomness that makes success predictions for individual papers very hard. However, we see our paper as a contribution to a novel and robust quantitative framework that can be used to build new incentive schemes for science production.

The presented approach has obvious shortcomings. The most striking is that papers are not classified by experts, neither as being mainstream or interdisciplinary, nor their quality in terms of being breakthrough or mediocre. The rewards studied to demonstrate that the BC network is indeed a useful concept for thinking of mainstream and interdisciplinarity, is itself still based on numbers of citations and rates thereof. A technical problem is the use of k-means clustering that we need for defining cluster centers. It is well possible that k-means clustering of the adjacency matrix of the BC network is too naive an approach. However, the fact that a similar effect is visible in the betweenness, even though smaller, indicates validity of the approach.

In conclusion, we think that in order to make science more than a self-sustained academic exercise and to avoid the dangers of being seen by the public and decision makers as a mere pastime of academics, it is paramount to change the current incentive scheme for science and research. To avoid the reported convergence towards mainstream it is necessary to think of how to reward authors in ways that incentivize out-of-the-box thinking, interdisciplinarity, and of course, actual problem-solving. A metric for such a reward scheme could indeed include the distance to clusters, measures of betweenness, and the degree of the BC network. It is conceivable that authors will try to optimize such schemes by using particular citing strategies and without producing more content. However, it would incentivize them to keep an open eye for developments in other areas of science other than their own.

## Data and methods

### Data

The American Physical Society (APS) data set used here includes 6, 040, 030 citations in all APS journal papers (Physical Review) published between 1893 and 2013 [27]. Besides citations, metadata records for 541, 448 papers over the same time period are available. Each record includes the digital objective identifier (doi), title, author(s), affiliation(s), publication date, and PACS numbers (if available).

### Bibliographic coupling network

In 1991 there were 9688 papers published in all the Physical Review journals A, B, C, D, Letters, and Reviews of Modern Physics [27]. Papers are uniquely identified by their digital objective identifier (doi). After removing editorials and errata (as provided in the meta information file of the APS data) 8831 papers remain. From these we construct the *bibliographic coupling (BC) network*, $M$, where paper $i$ is linked with paper $j$ if they both cite at least one common paper that was published before 1991. The weight on the (undirected) link, $M_{ij}$, is the overlap of the reference lists of paper $i$ and $j$. If both papers do not cite any third paper in common, $M_{ij}$ = 0. Nodes that are not linked to the largest connected component are excluded. The resulting BC network is finally composed of 8673 nodes and undirected 235, 971 weighted links. The BC network does not change with the arrival of new papers and their citations. Note that BC networks are very different from co-citation networks [28]. We identify 62, 266 authors in the author lists of the considered 8831 papers. We do not distinguish between authors and large collaborations that are identified as such.

### Characteristics of papers

We record the number of citations of every paper $i$ after two, $C_i^2$, ten, $C_i^{10}$, and twenty, $C_i^{20}$, years after its publication in 1991. The number of references cited in every paper is denoted by $L_i$. For every paper that appears in the BC network we compute the following properties. Weighted betweenness, $B_i = \sum_{s,t \in V} \frac{\sigma(s,t|v)}{\sigma(s,t)}$, where $V$ is the set of nodes, $\sigma(s, t)$ is the number of weighted shortest paths between nodes $s$ and $t$, and $\sigma(s, t|v)$ is the number of those paths going through node $v$. Weighted closeness, $K_i = (\sum_j d_{ij})^{-1}$, where $d_{ij}$ is the weighted network distance from node $i$ to $j$. The diversity of a paper we quantify by its PACS entropy: for every paper $i$ we construct the list, $PC_i$, of all PACS codes that appear in all the papers listed in the references of paper $i$. We then calculate the Shannon entropy of $PC_i$ as $I_i = -\sum_\alpha p_i^\alpha \log_2 p_i^\alpha$, where $p_i^\alpha$ is the (normalized) frequency of the PACS code, $\alpha$, in the list, $PC_i$. Not all papers have PACS information. To compute $I_i$ we only take papers for which there is PACS information for more than 80% of its cited references. Different thresholds were tested; results are very similar. Only 2491 papers meet the 80% criterion. To measure the distance, $D_i$, of paper $i$ to its nearest cluster center we use k-means clustering with a Hamming distance. $D_i = \min_\ell\{||\text{cluster}_\ell - \text{position}_i||_h\}$, where $\text{cluster}_\ell$ is the position of the center of cluster $\ell$, and $\text{position}_i$ is the position of node $i$; We chose $k$ = 20 clusters. All reported results are qualitatively very similar when 100 clusters are used. Because of their non-normality, we tested whether the medians of the distribution of $D_i$ changed over time with a two-sided Wilcoxon rank sum test.

## Supporting information

**S1 Text. Supplementary information to: The role of mainstreamness and interdisciplinarity for the relevance of scientific papers.**
(PDF)

## Author Contributions

**Conceptualization:** Stefan Thurner, Wenyuan Liu.

**Data curation:** Wenyuan Liu, Peter Klimek, Siew Ann Cheong.

**Formal analysis:** Stefan Thurner, Peter Klimek.

**Funding acquisition:** Stefan Thurner, Siew Ann Cheong.

**Investigation:** Stefan Thurner, Wenyuan Liu.

**Software:** Wenyuan Liu.

**Supervision:** Stefan Thurner.

**Validation:** Peter Klimek.

**Writing – original draft:** Stefan Thurner.

**Writing – review & editing:** Stefan Thurner, Wenyuan Liu, Peter Klimek, Siew Ann Cheong.

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
