## [Decision Letter · Decision Letter 0]

5 Feb 2020

PONE-D-20-01624

The role of mainstreamness and interdisciplinarity for the relevance of scientific papers

PLOS ONE

Dear Professor Thurner,

Thank you for submitting your manuscript to PLOS ONE. After careful consideration, we feel that it has merit but does not fully meet PLOS ONE’s publication criteria as it currently stands. Therefore, we invite you to submit a revised version of the manuscript that addresses the points raised during the review process.

This is really an interesting and valuable contribution. Both reviewers pointed to several minor issues, which are mostly misprints. These should be corrected before the manuscript can be accepted for publication. 

We would appreciate receiving your revised manuscript by Mar 21 2020 11:59PM. To enhance the reproducibility of your results, we recommend that if applicable you deposit your laboratory protocols in protocols.io, where a protocol can be assigned its own identifier (DOI) such that it can be cited independently in the future. For instructions see: http://journals.plos.org/plosone/s/submission-guidelines#loc-laboratory-protocols

We look forward to receiving your revised manuscript.

Kind regards,

Wolfgang Glanzel, PhD

Academic Editor

PLOS ONE

Journal Requirements:

2. Please amend your Data availability statement to indicate how other researchers can access the same datasets, for instance by providing contact details to the data holder or direct links. For more information on PLOS ONE data availability policies, please see https://journals.plos.org/plosone/s/data-availability.

Reviewers' comments:

Reviewer's Responses to Questions

**Comments to the Author**

1. Is the manuscript technically sound, and do the data support the conclusions?

Reviewer #1: Yes

Reviewer #2: Yes

2. Has the statistical analysis been performed appropriately and rigorously? 

Reviewer #1: Yes

Reviewer #2: Yes

3. Have the authors made all data underlying the findings in their manuscript fully available?

Reviewer #1: Yes

Reviewer #2: Yes

4. Is the manuscript presented in an intelligible fashion and written in standard English?

Reviewer #1: Yes

Reviewer #2: Yes

5. Review Comments to the Author

Reviewer #1: In this manuscript authors explore the bibliographic coupling network of the APS dataset in order to (1) identify mainstream, interdisciplinary and out-of-the-box scientific papers and (2) evaluate their performance over the past decades in terms of either absolute citations or citation rates. The paper is well written and well organized, the bibliography is exhaustive, the statistical analysis is rigorous and the results are interesting, convincing and well presented. Finally, the Data and Methods section and the Supporting Information material effectively contribute to clarify the procedures adopted. For all these reasons, in my opinion, the paper deserved publication on PLOSONE also in its present form.

Minor points:

-Figs.3(a) and (b) are not easily readable (in particular Figure 3(a)), since different green scales tend to confuse the reader; maybe it could be better to use just colored line profiles instead of filled boxes.

-References to the SI figures cited in the main text are missed (maybe it is a problem of latex compilation).

-In Ref.[7] the year is missed.

Reviewer #2: A remark: I think that interdisciplinary articles can be mainstream as well as out-of-the-box.

Interdisciplinarity is defined through the use of several PACS code, through weighted betweenness and also as being intermediate between clusters. Do these definitions coincide?

How is the center of a cluster defined? And how the periphery?

Page 2 line 54. It is not true that in general 67% of results in science cannot be reproduced. Please qualify this statement (and add reference).

Page 3 line 58. Write h-index (not h-factor)

Page 3 The authors discuss reasons why most scientists prefer incremental research. Yet, they do not mention that only few have the intellectual capacity to tackle the hard problems (most of us are not Einsteins).

Page 6 line 143 typo: interdiciplinary

Some references to figs in the SI are missing (??)

Page 9 line 275 Science of science is not a new field (although there is a revival of this term nowadays). Price already used this term and the term remained popular in China over the years.

Page 10 line 297 typo “fir” must be “for”

6. PLOS authors have the option to publish the peer review history of their article (what does this mean?). If published, this will include your full peer review and any attached files.

Reviewer #1: No

Reviewer #2: No

---

## [Author Response · Author response to Decision Letter 0]

19 Feb 2020

Data are owned by a third-party organization, the American Physics Society. We can not directly share the data with other researchers. However, APS make the data available for research, under data-requests@aps.org, however one must request access on their webpage, https://journals.aps.org/datasets.

---

## [Decision Letter · Decision Letter 1]

27 Feb 2020

The role of mainstreamness and interdisciplinarity for the relevance of scientific papers

PONE-D-20-01624R1

Dear Dr. Thurner,

We are pleased to inform you that your manuscript has been judged scientifically suitable for publication and will be formally accepted for publication once it complies with all outstanding technical requirements.

With kind regards,

Wolfgang Glanzel, PhD

Academic Editor

PLOS ONE

Additional Editor Comments (optional):

Reviewers' comments:

Reviewer's Responses to Questions

**Comments to the Author**

1. If the authors have adequately addressed your comments raised in a previous round of review and you feel that this manuscript is now acceptable for publication, you may indicate that here to bypass the “Comments to the Author” section, enter your conflict of interest statement in the “Confidential to Editor” section, and submit your "Accept" recommendation.

Reviewer #2: All comments have been addressed

2. Is the manuscript technically sound, and do the data support the conclusions?

Reviewer #2: Yes

3. Has the statistical analysis been performed appropriately and rigorously? 

Reviewer #2: Yes

4. Have the authors made all data underlying the findings in their manuscript fully available?

Reviewer #2: Yes

5. Is the manuscript presented in an intelligible fashion and written in standard English?

Reviewer #2: Yes

6. Review Comments to the Author

Reviewer #2: (No Response)

7. PLOS authors have the option to publish the peer review history of their article (what does this mean?). If published, this will include your full peer review and any attached files.

Reviewer #2: No

---

## [Editor Report · Acceptance letter]

6 Mar 2020

PONE-D-20-01624R1 

The role of mainstreamness and interdisciplinarity for the relevance of scientific papers 

Dear Dr. Thurner:

I am pleased to inform you that your manuscript has been deemed suitable for publication in PLOS ONE. Congratulations! Your manuscript is now with our production department. 

With kind regards,

on behalf of

Prof. Dr. Wolfgang Glanzel 

Academic Editor

PLOS ONE